# Chemical composition, seasonal variability and source identification of PM$_{2.5}$ in an urban background site in Medellín—Colombia

**Mauricio A. Correa-Ochoa**[1]*, **Miriam Gómez-Marín**[2], **Kelly Viviana Patiño-López**[2], **Luisa M. Gómez Pelaez**[1], **Santiago A. Franco**[1]

**1** Grupo de Investigación y Laboratorio de Monitoreo Ambiental—G-LIMA, Universidad de Antioquia—UdeA, Medellín, Colombia, **2** Grupo GHYGAM, Facultad de Ingeniería, Politécnico Colombiano Jaime Isaza Cadavid, Medellín, Colombia

\* mandres.correa@udea.edu.co

## Abstract

This study assessed the chemical characterization of PM$_{2.5}$ at an urban background site in Medellín—Colombia, a city situated in a topographically constrained valley exposed to air pollution accumulation influenced by its complex Andean topography and a combination of local and regional emission sources. A total of 112 samples of PM$_{2.5}$ were collected between March 2019 and March 2020, samples went through comprehensive sampling and chemical characterization (ICP-MS for metals, ion chromatography for anions, thermal/optical analysis for carbonaceous species). Daily PM$_{2.5}$ concentrations ranged from 8.15 to 37.86 $\mu$g/m³ (mean: 21.73±6.75 $\mu$g/m³). Principal Component Analysis (PCA) identified five major pollution sources, with mineral and resuspended dust (33.6% of explained variance) and secondary aerosols (17.0%) being the most prominent. The study area experiences clearly defined dry and wet periods, marked by distinct precipitation regimes. During the year these atmospheric conditions influence the concentration levels of pollutants. The integration of NOAA HYSPLIT back-trajectories and NASA FIRMS (VIIRS J2) fire hotspot data revealed long-range transport from the Magdalena, Orinoco and Amazon basins, deteriorating local air quality specially during dry periods. While local traffic and industrial emissions constitute a constant baseline, regional biomass burning and unfavorable meteorological conditions are the primary drivers of episodic high-pollution events. The study underscores the need for targeted strategies addressing both persistent sources as traffic or industrial emissions and episodic events to mitigate health and environmental impacts.

## Introduction

Urban environments are particularly affected by poor air quality [1]. The presence of atmospheric pollutants is due to local [2] and long-range transport sources [3]. PM$_{2.5}$

**Data availability statement:** All relevant data are within the paper and its Supporting information files.

**Funding:** This research was funded by Ecopetrol (grant no. DHS 118) the Area Metropolitana del Valle de Aburrá (grant number 787).

**Competing interests:** The authors have declared that no competing interests exist.

(particles with a diameter of less than 2.5 $\mu$m) is an atmospheric pollutant, that has attracted considerable interest worldwide due to its implications for public health and the environment [4]. $PM_{2.5}$ can travel long distances [5] and undergo transformation through photochemical reactions or interactions with other compounds present in the atmosphere [6]. The origin of $PM_{2.5}$ particles can be classified as primary, when emissions of the pollutant occur directly from the emission source, or secondary, when the pollutant is formed through from reactions that occur in the atmosphere [7].

These particles are microscopic, which allows them to penetrate the lungs and remain deposited there [8]. They can also enter the bloodstream, leading to respiratory and cardiovascular diseases [9]. The relationship between this pollutant and other types of conditions has been studied. Some examples include cognitive decline [10], damage to epithelial cells [11], and growth deficits [12]. This pollutant has also been identified as a potential threat to ecological systems [13]. The inhalation of these particles by animals can cause adverse effects [14], and bioaccumulation within plant tissues has also been observed [15]. On the other hand, $PM_{2.5}$ can affect the climate and hydrology [16]. It can alter the amount of solar radiation that reaches the earth's surface by changing the distribution of solar energy in the atmosphere and by modifying the hygroscopic growth of cloud condensation nuclei [17]. $PM_{2.5}$ is the main cause of smog, which reduces visibility in cities [18].

Economic and industrial development is linked to increased emissions of air pollutants [19,20]. PM consists of a heterogeneous mixture of materials emitted from both anthropogenic (e.g., mining, agriculture, industry, transportation, and construction) and natural (e.g., the resuspension of crustal material, volcanic eruptions, and suspension of sea salts) sources [21,22]. Forest fires and biomass burning are also significant sources of this pollutant. Due to the diversity of sources, the pollutant's composition is heterogeneous [23]. However, studies have determined that the composition of PM is primarily composed of ions, metals, minerals [24], and carbon compounds [25], among other components [26].

Each group of compounds has specific implications. For instance, high concentrations of heavy metals are linked to diseases like cancer [27]. Regarding anions, sulfate is associated with environmental issues such as acid rain [28], while areas affected by nitrate pollution are often characterized by reddish skies [29]. Carbonaceous species play an important role in the climate balance as they contain particles that absorb large amounts of energy [30]. Organic carbon (OC) contains a variety of aliphatic, aromatic, and acidic compounds, those are compounds with a complex chemical composition containing numerous toxic substances, such as polycyclic aromatic hydrocarbons (PAHs), which are linked to respiratory and cardiovascular diseases [31,32].

Medellín, located in a narrow valley in the Colombian Andes, is particularly exposed to $PM_{2.5}$. Rapid population growth and urbanization have led to increased air pollution and have affected the quality of life of its inhabitants. Recent studies have examined the health effects of short and long-term exposure to air pollutants, as well as the local incidence of diseases caused by $PM_{2.5}$ exposure [33]. These studies have found a relationship between $PM_{2.5}$ exposure and acute and chronic respiratory and cardiovascular conditions, as well as neoplasms [34]. $PM_{2.5}$ has also been found to have both cytotoxic

and genotoxic potential [35]. An estimated 553 deaths per year in Medellín are attributable to PM$_{2.5}$ exposure [33,36]. It also has also been estimated that PM$_{2.5}$-related deaths could increase by over 150% between 2016 and 2030 [37].

Previous studies have analyzed the PM$_{2.5}$ problem in Medellín from different perspectives. These studies have used predictive models to estimate the pollutant's impact [37,38], analyzed the influence of specific sources [21], used spatial distribution models [39], and performed chemical characterizations to determine its components and emission sources [4,35,40]. However, there are few studies that have examined the relationships between emission sources and how pollutants behave throughout the year, considering the city's meteorological variations.

The chemical characterization of PM$_{2.5}$ samples includes the analysis of metals (using inductively coupled plasma mass spectrometry), ions (using ionic chromatography), and carbonaceous species (using thermo-optical methods). The sources of the pollutants were determined using principal component analysis (PCA). Samples were collected over the course of a year, taking into account the dry and wet seasons characteristic of the study region. This study characterizes PM$_{2.5}$ particulate matter in an urban area of the city of Medellín, identifies the main emission sources, and compares pollutant concentrations during dry and wet periods in relation to regional climatic factors such as precipitation and wildfires reported both within the local urban domain and in other regions of the country. This approach improves understanding of how emissions, climate, and regional transport jointly control PM$_{2.5}$ variability in complex urban environments. It provides evidence to design season-specific air quality strategies and to anticipate pollution episodes linked to climate variability and regional fires.

## Methodology

### Study area and sampling site

Medellín is the second-largest city in Colombia. It is located in a narrow valley (Aburrá Valley) which is characterized by frequent thermal inversions and limited dispersion of pollutants [35]. The average temperature is 21.5 °C; the average total annual rainfall is 1685 mm. During the year, there are two dry seasons and two rainy seasons. The rainy seasons extend from the end of March to the beginning of June and from the end of September to the beginning of December [41].

The monitoring site is located at the Pedro Justo Berrio Institute in an urban area with densely populated characteristics (latitude: 6.243; longitude: −75.612; elevation: 1,614 m), close to the meteorological station Pedro Justo Berrio. The sampling point is on a 3-meter-high terrace more than 50 meters from major paved roads. Near the sampling point, there are quarries extracting materials for the brick industry, so samples captured at the station contain a mixture of urban emission sources. Fig 1 shows the location of the sampling point.

### Sampling design and filter collection

Sample collection was performed following standardized procedures validated by the GHYGAM Laboratory of the Universidad Politécnica Colombiana Jaime Isaza Cadavid, in accordance with the Quality Assurance Instructions ID-MEA183 and the Sampling Instructions of its Management System, which is accredited according to ISO/IEC 17025 [42]. PM$_{2.5}$ samples were collected using low-volume equipment (PQ200 by BGI Instruments®, flow rate: 16.67 L/min) with 47-mm-diameter Teflon filters, according to international reference methods (CFR 40, Appendix L to Part 50, Reference Method for Determining Fine Particulate Matter as PM$_{2.5}$ in the Atmosphere, [43]).

Before sampling, each Teflon filter was weighed three times, with an average difference of no more than 15 $\mu$g, using a microbalance with a resolution of 0.001 mg (Sartorius CPA26P). Prior to gravimetric analysis, filters were conditioned in a desiccator under controlled temperature and relative humidity before and after sampling to remove residual moisture and ensure mass stability, thereby reducing hygroscopic effects on PM$_{2.5}$ determination. All samplings underwent flow calibration, and environmental conditions were recorded.

Sampling times complied with the parameters described in the Protocol for Air Quality Monitoring [44], lasting 23 ± 1 hours between 12:00 a.m. and 12:00 p.m. A total of n = 112 samples were collected between April 2019 and March 2020

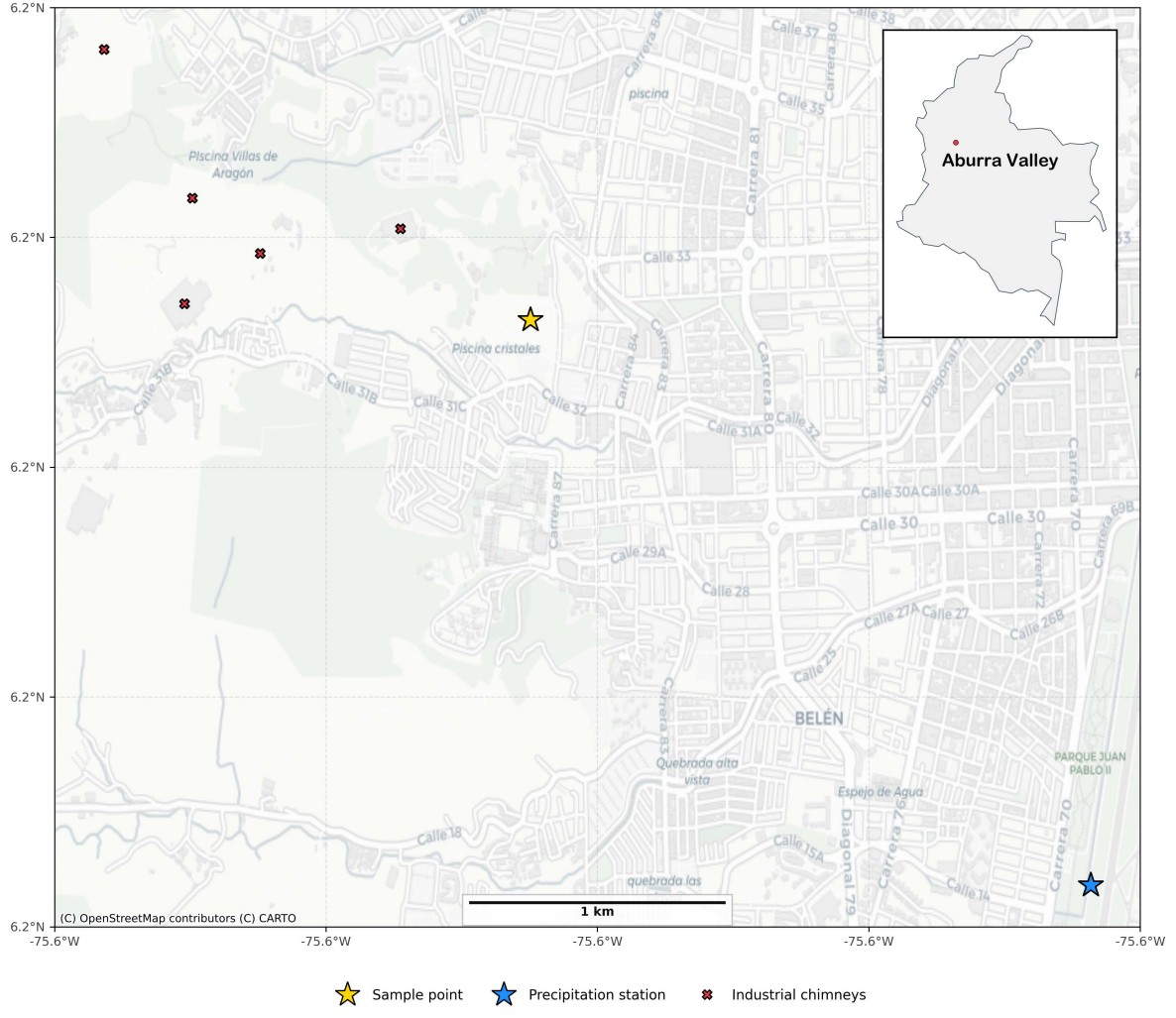

**Fig 1. Sampling point location in the urban area of Medellín.** Map base layer from OpenStreetMap (© OpenStreetMap contributors). The figure illustrates the geographical placement of the monitoring station at the Pedro Justo Berrio Institute.

to determine PM$_{2.5}$ concentrations and perform chemical composition analyses. The study period covers the biennial cycle of annual rainfall in Medellín; Table 1 shows the periods and the number of samples collected in each period.

After sampling, the PM$_{2.5}$ samples are transported to the laboratory under the cold chain to ensure a temperature of 4 °C or lower during transport. The samples are also protected from light to prevent the loss of volatile and photosensitive compounds. Once in the laboratory, the samples underwent final temperature and humidity conditioning for a minimum of 24 hours. According to the volumes of filtered air for each sample, the concentration of PM$_{2.5}$ in ambient air is calculated. Then, the samples were stored at −20 ± 3 °C until chemical characterization.

## Chemical characterization

**Metals by inductively coupled plasma mass spectrometry (ICP-MS).** The content of metals was determined by ICP-MS in a Thermo Scientific instrument, model I CAP-RQ, using reference protocols and methods [45,46]. The technique is based on the coupling of a method to generate ions (inductively coupled plasma) and a method to separate

**Table 1. Samples by period collected between 2019 and 2020.**

| Season (Year) | Period | Samples (n) |
|---|---|---|
| Wet 1 (2019) | March — May | 25 |
| Dry 1 (2019) | June — August | 31 |
| Wet 2 (2019) | September — November | 30 |
| Dry 2 (2019–2020) | December — Febraury | 26 |

and detect them (mass spectrometer). The sample material in solution is introduced by pneumatic nebulization into a radio-frequency produced plasma where energy transfer processes cause desolvation, atomization, and then ionization. The ions are extracted from the plasma through a pressure interface, produced by a vacuum pump, and separated based on their mass-to-charge ratio by a quadrupole mass spectrometer, which has a minimum capacity resolution of 1 amu (atomic mass unit) at 5% of the peak height. The ions transmitted through the quadrupole are detected by an electron multiplier detector, or by a Faraday detector, and the resulting information is processed by a data management system.

**Ions by ionic chromatography.** The concentration of water-soluble ions ($K^+$, $Na^+$, $Mg^{2+}$, $Ca^{2+}$, $SO_4^{2-}$, $NO_3^-$, $F^-$ and $Cl^-$) in $PM_{2.5}$ is determined by Ion Chromatography using the reference method [47] in Thermo Scientific Dionex Aquion reference equipment. A sample of the particulate material extraction in water is injected into a stream of carbonate-bicarbonate eluent for anions and metasulfonic acid for cations and passed through a series of ion exchangers where the ions of interest are separated according to their relative affinities and subsequently identified based on the basis of retention time compared to standard solutions of each ion. Quantitative determination is performed by measuring the peak area of each ion.

**Carbonaceous species by thermo-optical methods.** Organic Carbon (OC) and Elemental Carbon (EC) are analyzed over temperature ramps in oxidizing and/or non-oxidizing atmospheres, using Thermal/Optical Reflectance (TOR) and Thermal/Optical Tramitance (TOT) methods. This analysis consists of heating a portion of the sample continuously with the detection of volatilized carbon or oxidized carbon coming out of the sample. With consistent standardization, these methods provide equivalent measurements of Total Carbon (TC) donde $TC = EC + OC$ and define the split between OC and EC and their fractions, based on combinations of combustion temperatures, residence time at each temperature, the composition of the atmosphere surrounding the sample, and the light reflected or transmitted through the filter.

Organic and elemental carbon are determined by the NIOSH 5040 method, applying the experimental conditions established in the protocol [48] and using the Sunset thermo-optical meter, reference 5L. The principle of measurement consists of differentiating EC and OC in atmospheric particles collected and the detection of methane produced by both fractions [49].

## Meteorological context and regional transport assessment

To characterize the seasonal variability of the study area, precipitation data were obtained from the Institute of Hydrology, Meteorology, and Environmental Studies (IDEAM), specifically from the station located at the Olaya Herrera Airport (6.22° N; −75.59° W). Complementary to the local meteorology, the influence of regional biomass burning emissions was evaluated using active fire data retrieved from the NASA Fire Information for Resource Management System (FIRMS) [50].

We utilized the VIIRS sensor aboard the NOAA-20 satellite (product J2_VIIRS_C2, 375 m spatial resolution) to identify hotspots across potential source regions during the studied period, including the Magdalena, Orinoco and Amazon basins. These data were spatially filtered and aggregated into daily counts to assess their relationship with biomass burning tracers detected at the sampling site. Finally, to determine the origins of air masses arriving at the Aburrá Valley, backward trajectories were computed using the NOAA HYSPLIT model with GDAS1 meteorological data (1° resolution) [51,52]. Trajectories were calculated for a duration of 96 h backwards in time at [12:00 UTC], setting arrival heights at [500, 1000, and 1500] m above ground level (AGL) to capture transport dynamics within and above the planetary boundary layer.

## Data and source analysis

A descriptive statistical analysis was performed. Subsequently, the data for each period was analyzed in order to reconstruct the $PM_{2.5}$ mass. Principal component analysis (PCA) was performed to identify the sources of the pollutant, from the whole species-resolved dataset. Prior to the analysis, all the chemical species with more than 50% missing values were excluded, and the remaining concentrations below the detection limit were imputed as half of the limit.

Spearman correlation tests were then used to eliminate variables that did not exhibit at least moderate correlation ($\rho < |0.5|$, $p > 0.05$) with other species, ensuring that only interrelated tracers remained. The concentration data were then mean-centered and variance-scaled to standardize differences in magnitude and units between ions, metals, and carbonaceous species. Then, a PCA was performed on the resulting correlation matrix using the varimax rotation criterion to improve the interpretability of component loadings. Each rotated principal component (PC) was interpreted as a distinct emission source based on the pattern of highly loaded species. The number of components retained captured at least 70% of the total variance in the dataset. The procedures were performed using R Studio 2026.0, and Python 3.11.9 software. The data and procedures developed for the statistical procedures are available and replicable through the material attached to this work (S1 File)

## Results and analysis

The average concentration of $PM_{2.5}$ was $21.73 \pm 6.74$ $\mu$g/m³ during the study period, with maximum (37.9 $\mu$g/m³) and minimum (8.1 $\mu$g/m³). Among the study periods, Dry 2 had the highest average concentration of the pollutant, while the lowest concentration was observed in the Dry 1 period. OC concentrations remained stable between periods, while EC concentrations were significantly lower in the dry periods. Table 2 shows a summary of the main species analyzed.

Table 3 shows the maximum, minimum, and average values in each study period for $PM_{2.5}$, where it can be seen that in the Dry 1 period there were the lowest concentration peaks and on average the lowest concentrations, contrasting with what happened in Dry 2, where the concentrations reached maximum peaks.

Fig 2 offers a comparison of the chemical compounds that were found most frequently in the samples characterized throughout each period, where it is clearly evident that OC is the main component of the $PM_{2.5}$ mass, followed by sulfate.

**Table 2. Average and standard deviation of the main analyzed species ($\mu$g/m³) over the studied periods.**

| Variable | General | Wet 1 | Dry 1 | Wet 2 | Dry 2 |
|---|---|---|---|---|---|
| $PM_{2.5}$ | 21.729±6.747 | 21.932±7.210 | 18.204±5.002 | 19.918±7.446 | 26.683±6.303 |
| Organic Carbon (OC) | 7.286±3.090 | 7.755±3.846 | 5.804±1.916 | 7.741±3.557 | 7.930±2.470 |
| Elemental Carbon (EC) | 0.896±0.469 | 1.095±0.537 | 0.652±0.212 | 1.167±0.561 | 0.724±0.213 |
| Sulfate ($SO_4^{2-}$) | 1.806±0.998 | 2.638±1.462 | 1.142±0.452 | 1.650±0.640 | 1.945±0.627 |
| Nitrate ($NO_3^-$) | 0.617±0.421 | 0.799±0.638 | 0.350±0.245 | 0.692±0.287 | 0.663±0.317 |
| Chloride ($Cl^-$) | 0.156±0.144 | 0.124±0.169 | 0.077±0.070 | 0.138±0.154 | 0.277±0.150 |
| Fluoride ($F^-$) | 0.021±0.018 | 0.014±0.009 | 0.022±0.015 | 0.021±0.013 | 0.029±0.029 |
| Sodium ($Na^+$) | 0.119±0.089 | 0.098±0.051 | 0.079±0.038 | 0.177±0.119 | – |
| Potassium ($K^+$) | 0.168±0.113 | 0.213±0.150 | 0.106±0.057 | 0.190±0.079 | – |
| Magnesium ($Mg^{2+}$) | 0.026±0.023 | 0.024±0.019 | 0.016±0.012 | 0.038±0.030 | – |
| Calcium ($Ca^{2+}$) | 0.268±0.183 | 0.263±0.203 | 0.201±0.128 | 0.339±0.193 | – |
| Iron (Fe) | 0.429±0.366 | 0.509±0.459 | 0.270±0.174 | 0.547±0.407 | 0.395±0.347 |
| Aluminum (Al) | 0.380±0.352 | 0.456±0.448 | 0.211±0.151 | 0.509±0.379 | 0.344±0.315 |

All concentrations are reported in $\mu$g/m³. Values represent the mean±one standard deviation. Cations ($Na^+$, $K^+$, $Mg^{2+}$, $Ca^{2+}$) were not available for the Dry 2 period.

**Table 3. Summary of PM$_{2.5}$ ($\mu$g/m³) during the study period.**

| Period | Min | Max | Mean |
|--------|-----|-----|------|
| Wet 1 | 11.16 | 37.09 | 21.93 |
| Dry 1 | 10.06 | 26.57 | 18.43 |
| Wet 2 | 8.15 | 37.86 | 19.64 |
| Dry 2 | 12.18 | 35.14 | 23.50 |

Minimum, maximum, and average PM$_{2.5}$ concentrations for the four meteorological periods in the Medellín urban area.

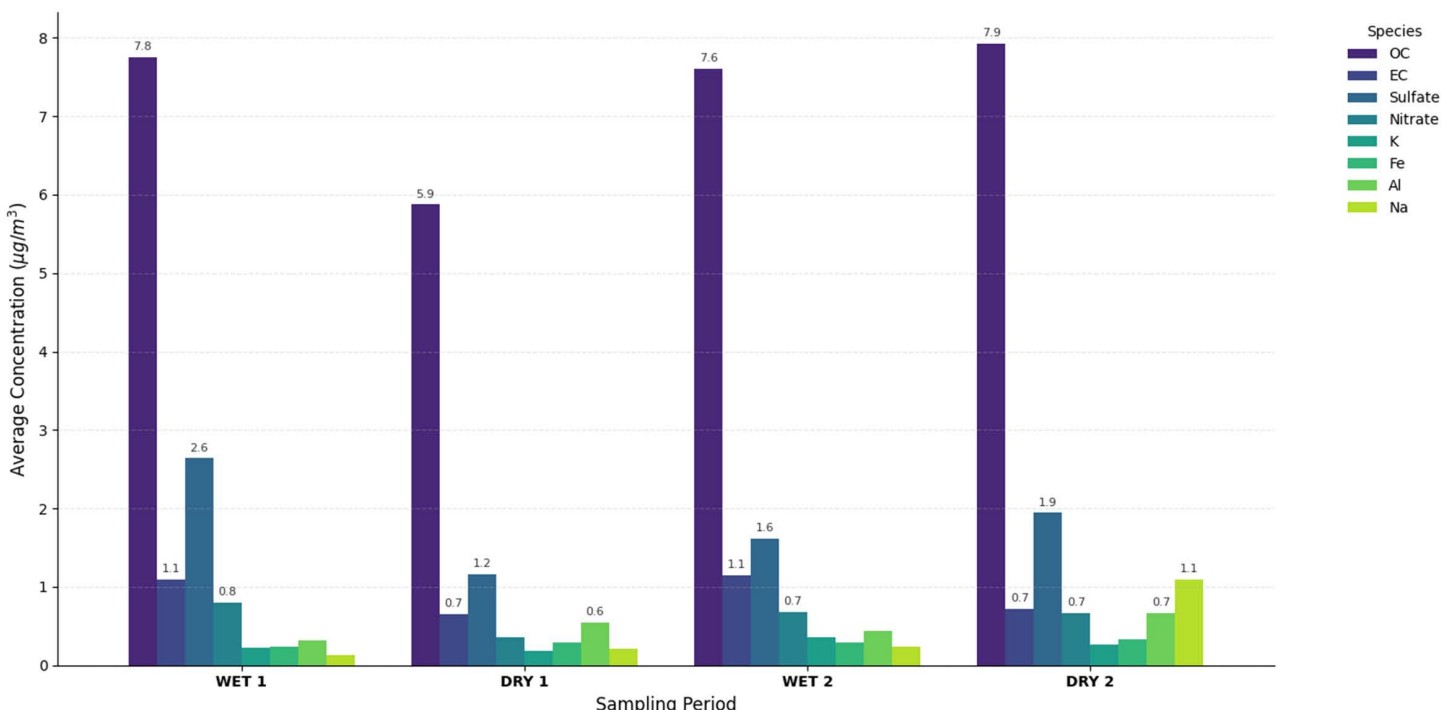

**Fig 2. Concentration ($\mu$g/m³) of the main species.** Comparison of the most frequent chemical compounds found in PM$_{2.5}$ samples across the four study periods. The chart illustrates that Organic Carbon (OC) is the primary component of the mass, followed by sulfate and other soluble ions.

During the study period, it is evident that the concentration of soluble ions constitutes most of the PM$_{2.5}$ mass after the carbonaceous species.

Among the metals analyzed, it was found that, on average, the concentrations of carbonaceous species and sulfate are the highest, followed by nitrate, Al, Na, Ca, Fe, K, Si, Mg, Zn, Ti, Ba, Cu, Pb, Mn, Ni, Cr, V, Sb, As, Se, Cd, Mo, Co, Hg, Ag and Be. Fig 3 shows the boxplot graphic for each period in which the differences between the magnitudes and variability of EC, OC, and PM$_{2.5}$ pollutants can be seen. OC has intermediate concentrations, with a median of approximately 6 $\mu$g/m³, but shows several significantly high outliers exceeding 12 $\mu$g/m³, while EC shows the lowest concentrations and the least variability, with a very narrow box and a median just above 1 $\mu$g/m³.

Fig 4 shows the temporal variation of carbonaceous compounds and PM$_{2.5}$ during the study period, where concentrations remain relatively stable, with peaks during dry and wet periods. The highest concentrations of the pollutant were obtained in the final phase of sampling.

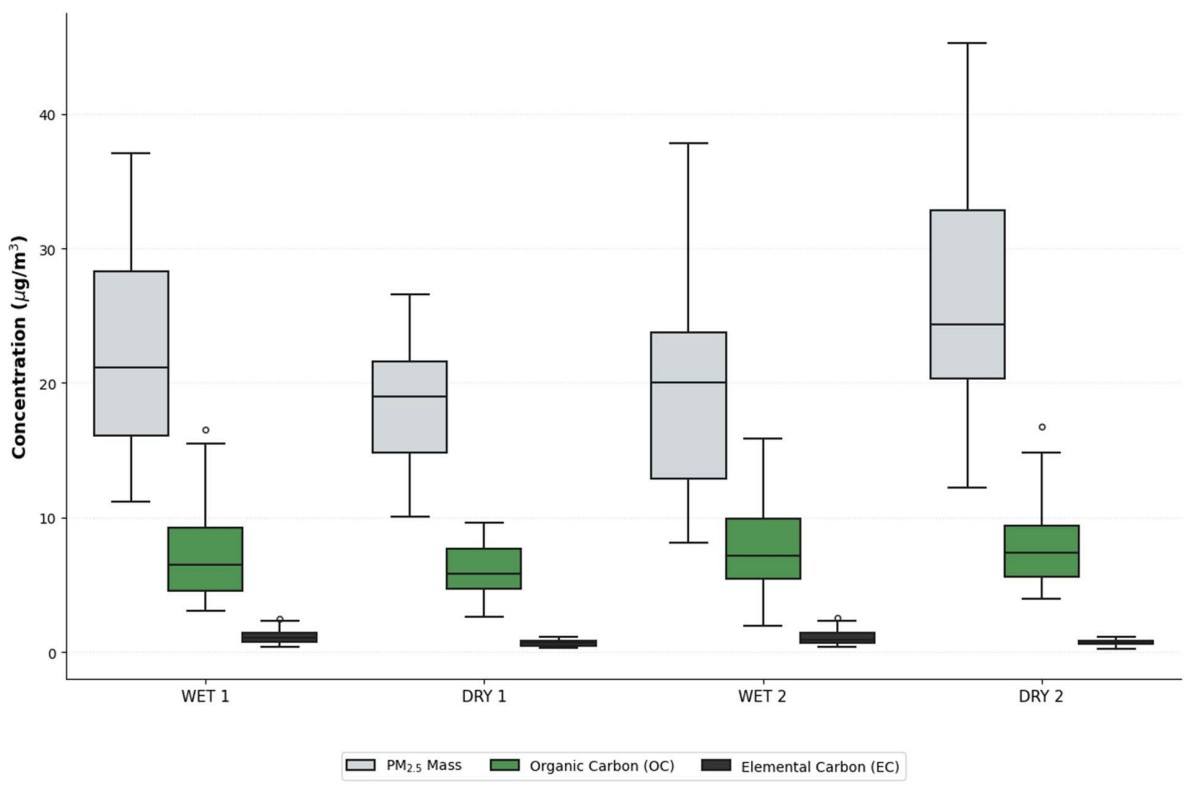

**Fig 3. Boxplot for PM$_{2.5}$, and OC and EC species for each period.** The figure illustrates the distribution, medians, and outliers for the major carbonaceous fractions and total PM$_{2.5}$ mass. Differences in magnitude and seasonal variability are highlighted for each study period.

In the Dry 2 period, there is a tendency for PM$_{2.5}$ concentrations to increase, the concentration peaks occurred during the wet periods, the OC and EC concentrations are closely aligned with PM$_{2.5}$ concentrations, presenting peaks of maximum values on similar dates, the OC/EC ratio had an average value above 6 throughout the study period, mainly associated with biomass burning [53,54]. However, concentrations vary greatly, and pollution peaks could be associated with activities outside the valley [55]. National regulations estimate a concentration below 37 $\mu$g/m$^3$ with an acceptable AQI. During the study period, only two samples were above this level both in the wet periods [44].

The study area delimited for the evaluation of hot spots [−78°W, −3°N; −64°W, 13°N] considering the Colombian territory and part of eastern Venezuela, shows the number of hot spots per km$^2$, in this case it can be seen that in the north of the continent there is usually a greater number of hot spots in the first half of the year and a notable reduction for the second half. Fig 5 shows the fires reported during the study period for the aforementioned area, where relationships can be found between the number of hot spots and PM$_{2.5}$ concentrations.

To gain a clearer picture of what is happening in the region, a cross-section was taken for the Aburrá Valley [−75.8°W, 6.0°N; −75.3°W, 6.5°N], where 141 occurrences were identified for the study period, concentrated mainly in the dry seasons. Fig 6 clearly shows how specific events coincide with peaks in PM$_{2.5}$ concentration in the region.

On the other hand, precipitation regimes in the city appear to be related to peak PM$_{2.5}$ concentrations during wet periods; Fig 7 shows the trend of precipitation records from the Olaya Herrera station, located near the study area.

The above information suggests that different conditions affecting air quality and particulate matter pollution exist in the study area. In dry periods, there is a strong presence of fires in the vicinity of the valley and in large basins of the country.

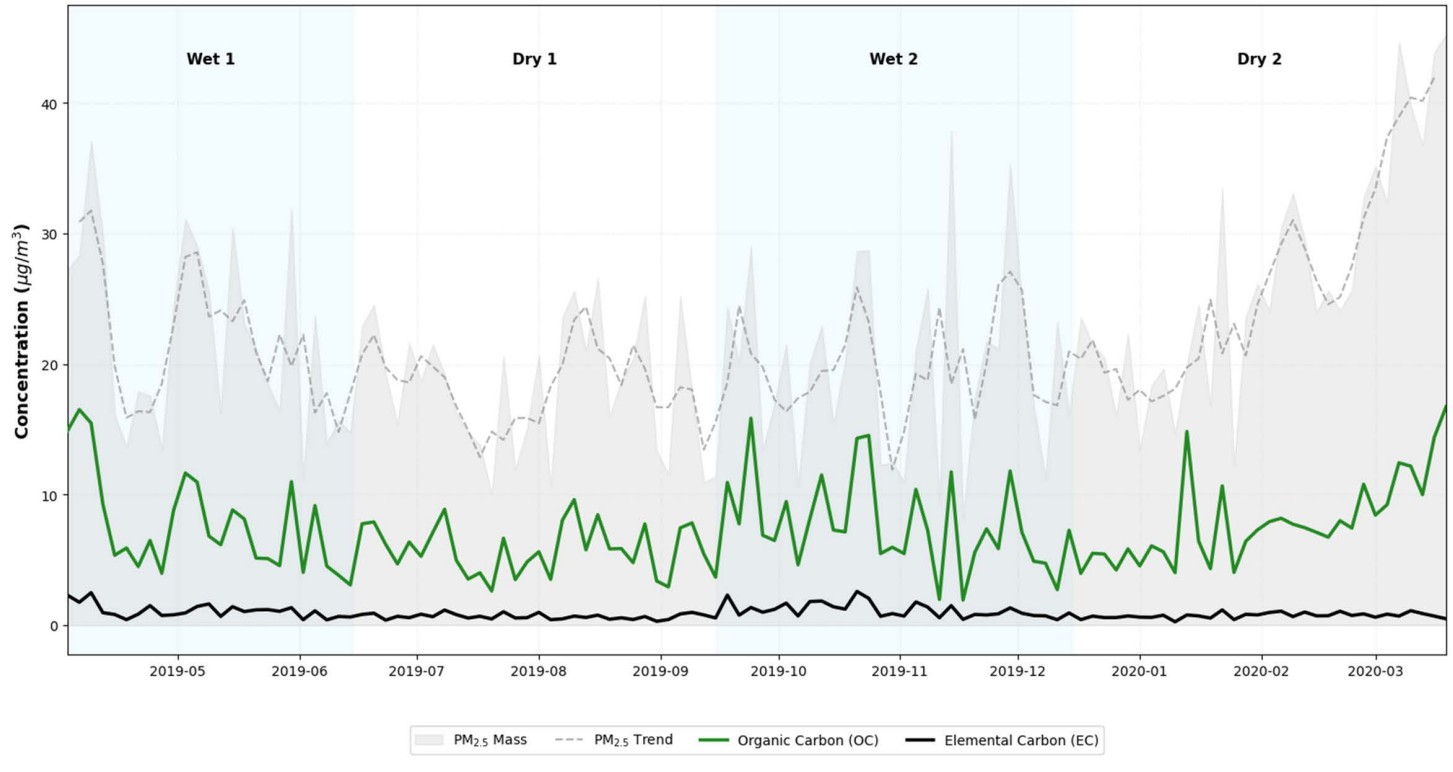

**Fig 4. Distribution of PM$_{2.5}$, OC and EC species during the study period.** Temporal variation of total mass and carbonaceous fractions. The series illustrates stable baseline levels with episodic peaks occurring across different meteorological seasons, reaching maximum values during the final sampling phase.

In wet periods, higher levels of precipitation and lower levels of solar radiation hinder the dispersion of pollutants. This is contrasted with Fig 8, which shows critical dates with pollution peaks (April 2019 Wet 1 period; September 2019 Dry 1 period; November 2019 Wet 2 period; March 2020 Dry 2 period).

At the first pollution peak, it is evident that the air currents do not specifically originate from areas of the country most affected by fires. However, during this period, there are high levels of precipitation at the beginning of September, coinciding with a lower density of hotspots and lower PM$_{2.5}$ concentration values reported during the study period. In November, when there is not a high density of hotspots in the region, the high pollutant concentrations could be attributed to the region's atmospheric conditions, whereas in March 2020, when the highest fire density peaks are in northern South America and the Aburrá Valley, pollutants are transported directly to the valley, particularly from the lower Magdalena River basin.

Although the arguments are based on the evidence provided by the different parameters, the anthropogenic emission characteristics of the city, which are mainly characterized by the vehicle fleet and industrial sector, cannot be ignored. The 2022 emissions inventory determined that mobile sources in Medellín emit slightly more than 3,500 ton/year of PM$_{2.5}$, while industrial sources emit approximately 160 ton/year. Therefore, the chemical characterization and determination of sources complement the presented analysis.

Table 4 shows the results of the correlation analysis where high ratios between species such as Al and Si are identified, indicating that their presence comes from the same source, usually attributable to crustal material [56,57]. The high

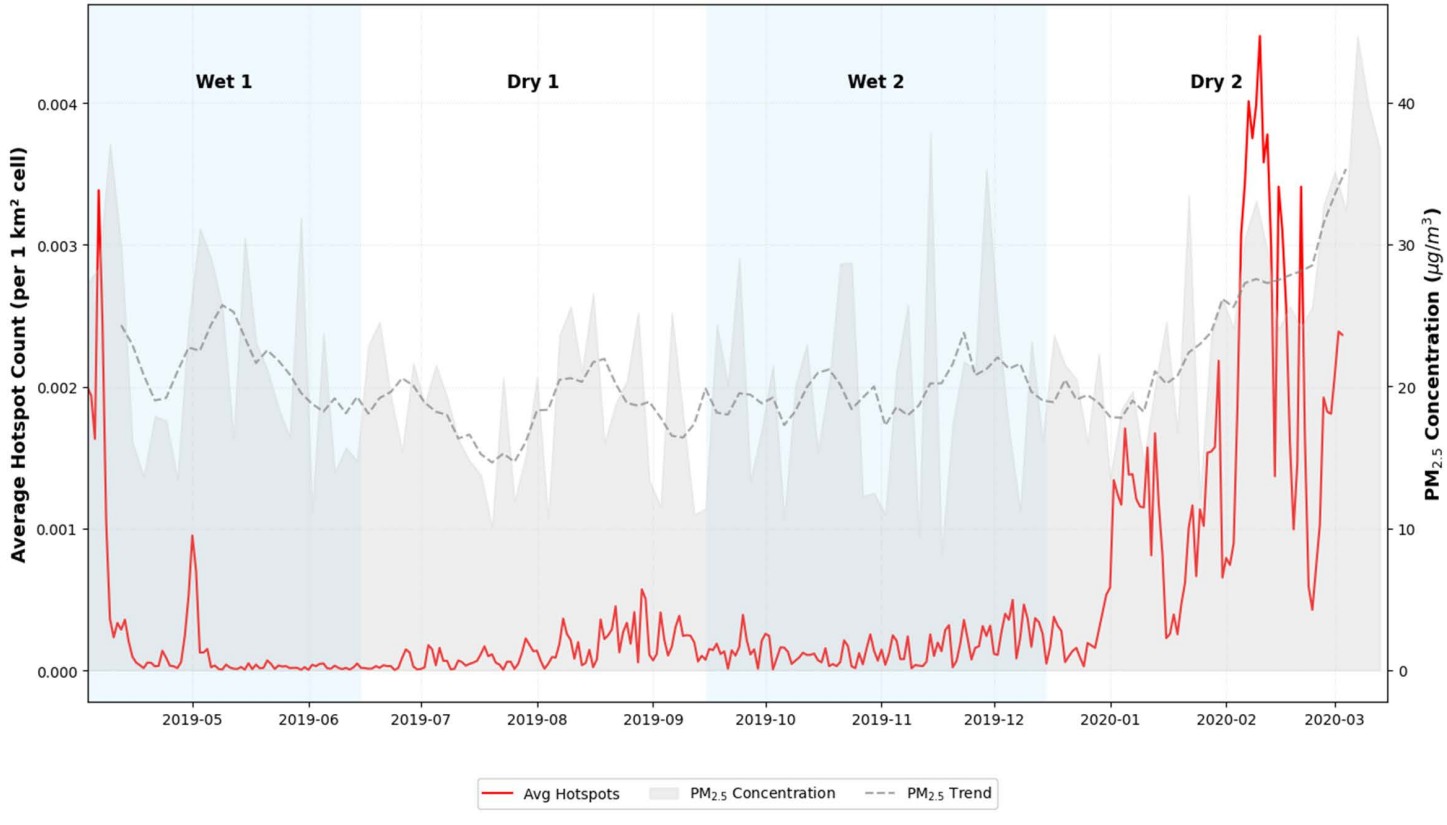

**Fig 5. Hotspots dynamics (North of South America) vs PM$_{2.5}$ concentration.** Temporal correlation between regional fire activity and local particulate matter levels. The figure illustrates the seasonal density of hotspots across Colombia and Venezuela in relation to the observed PM$_{2.5}$ peaks in Medellín.

correlation between PM$_{2.5}$ and OC (0.82) is related to the high carbonaceous species content in the pollutant, which averages 34% of the mass and reaches 78% in the Dry 2 period.

Table 5 shows the loading values of the PCA, grouping the variables into five principal components (PC) representing common emission sources accounting for more than 70% of the total variance. PC1 is dominated by elements of crustal or soil dust origin, with high loadings for Al, Si, Ca, Ti, Fe, Mg and Ba [58,59]. PC2 represents the source of mixed combustion and secondary aerosols, as it groups the PM$_{2.5}$ mass with its main carbonaceous components (OC, EC) and secondary anions ($NO_3^-$, $SO_4^{2-}$) [60,61]. PC3 represents a soluble ion factor, related to biomass burning (high $K^+$ loading) [62,63]. PC4 identifies an industrial or non-exhaust emission source, highlighted by high loadings of heavy metals such as Zn and Pb [64,65]. Finally, PC5 with $F^-$ and $Cl^-$ halogens, suggesting specific industrial sources [66,67].

Five different components explained 72% of the variance. PC scores were calculated for each sample to estimate the relative contribution of each inferred source to PM$_{2.5}$ mass. The resulting source profiles and time series of component scores provide a foundation for interpreting seasonal and episodic variations in urban PM$_{2.5}$ pollution.

The scree plot (see Fig 9) shows how much of the total variability in the dataset each principal component (PC) captures. The first component towers above the rest, explaining roughly one-third of the variance (33.6%). The second component adds another 17%, bringing just over half of the total. The third, fourth and fifth components each contribute progressively smaller—but still meaningful—portions: about 8.8%, 6.8% and 5.6%, respectively. A reference line drawn at 5 percent highlights that only the first five components rise above that threshold, suggesting they each carry at least the

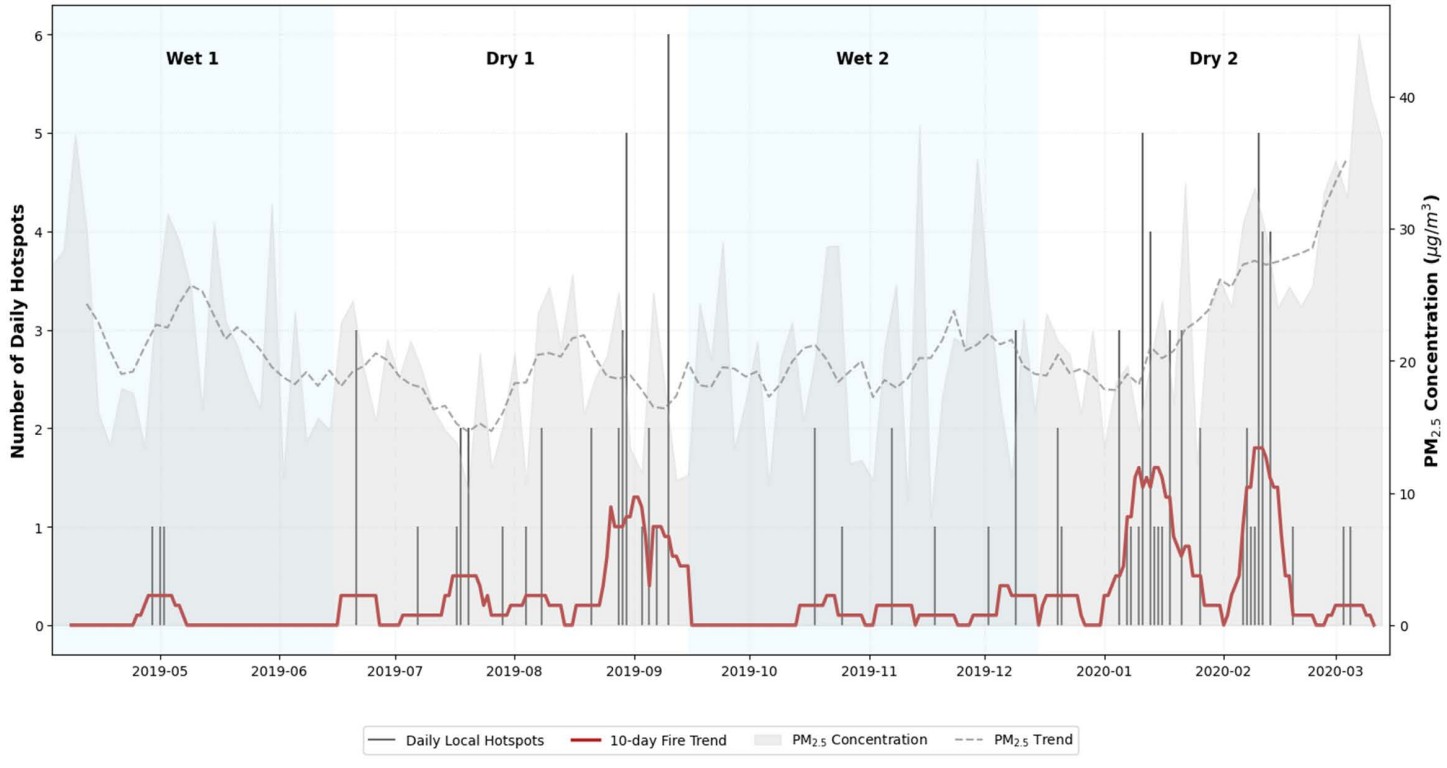

**Fig 6. Hotspots dynamics in the Aburrá Valley vs PM$_{2.5}$ concentrations.** The figure illustrates the temporal coincidence between localized fire events (hotspots) within the valley's cross-section and the highest recorded peaks of particulate matter, particularly during the dry periods.

equivalent of one original variable's worth of information. Beyond the fifth, every additional component explains less than 5% of the variance.

These five components capture the bulk of the meaningful structure in the data. After that, each extra dimension adds only small, incremental detail—likely representing finer patterns or statistical noise—so a five-component solution strikes a good balance between simplicity and fidelity.

## Discussion

The findings from this source apportionment study at the Belén urban background station in Medellín provide insights into PM$_{2.5}$ composition, seasonal dynamics, and emission sources. PCA results reveal five source profiles that collectively explain 71.8% of PM$_{2.5}$ variability, aligning with Medellín's unique geographical and anthropogenic context while offering actionable pathways for air quality management. Table 6 shows the resume of the source profiles.

PC1 explains 33.6% of the variance, where elements derived from the earth's crust are found. These compounds also have a greater presence in dry periods, which is associated with an increase in the resuspension of this type of materials under low humidity conditions [4]. The abrupt changes in the topography of the valley (mediated by the narrow valley characteristic) added to the daily activities of the valley such as construction, commerce and mobility.

For PC2, the influence of PM$_{2.5}$ associated with nitrates, sulfates, OC and EC add up to 17% of the sample variance. This factor includes the primary combustion sources of precursor aerosols and the formation of secondary species in the atmosphere, where high humidity and intense sunlight fuel the conversion of gases into particles. Previous studies in Medellín have noted similar trends, indicating that secondary aerosols, including those from biomass burning, drive about

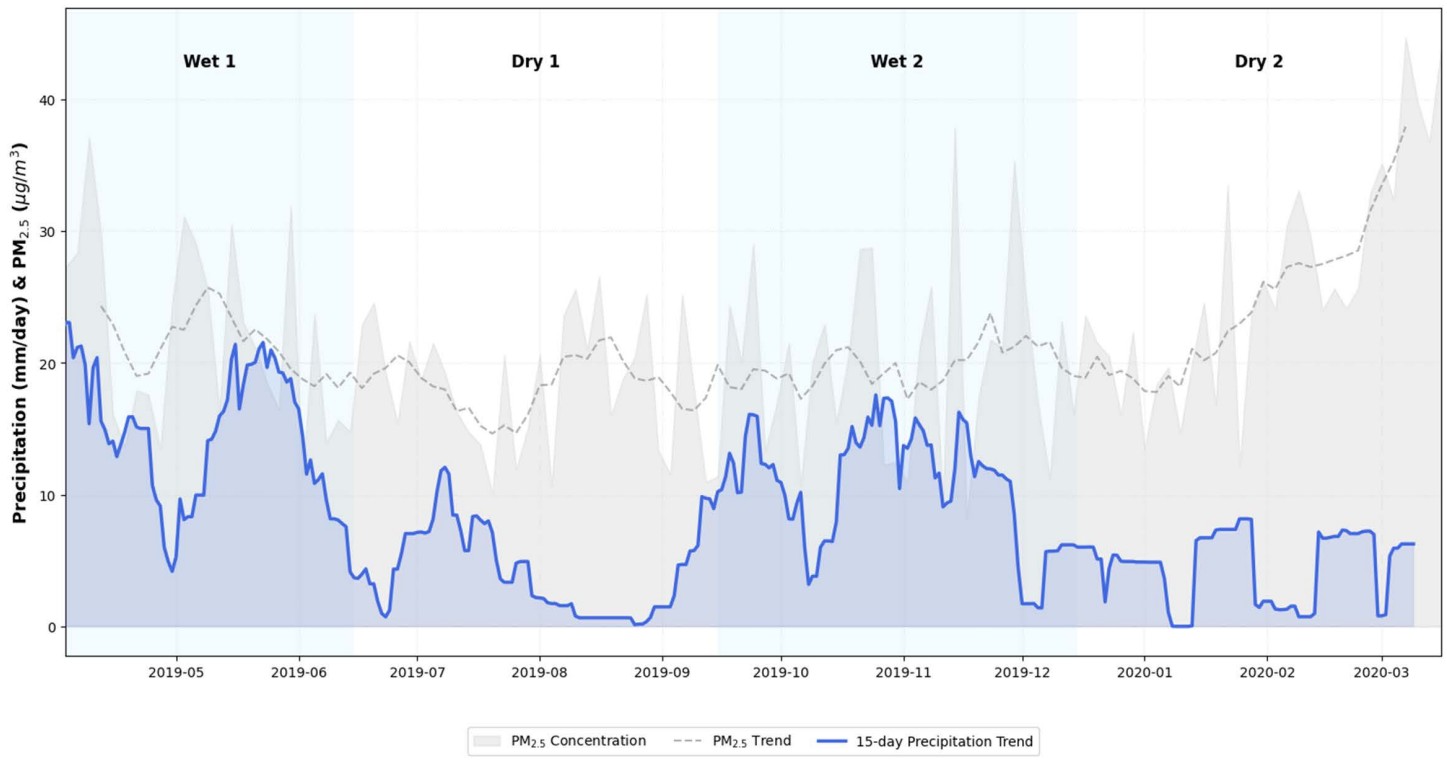

**Fig 7. Precipitation trend at Olaya Herrera's station vs PM$_{2.5}$ concentration.** Comparison between daily rainfall accumulation and particulate matter levels. The figure highlights how precipitation events influence the scavenging or accumulation of PM$_{2.5}$ during the transition between seasons.

22% of PM$_{2.5}$ levels. Biomass burning, such as agricultural fires, is a major source of secondary organic aerosols [40]. OC/EC ratios above 8 were observed during wet seasons, suggesting a mix of biomass combustion and traffic emissions.

PC3 (8.8% variance) identifies the potassium ion as an important indicator of biomass burning [68]. Although K concentrations peak in Dry 2 (0.38 $\mu$g/m$^3$), its weaker correlation with OC during the dry season ($\rho$ = 0.46) suggests distinct combustion phases: agricultural fires during dry periods versus fuel burning accumulation year-round [69].

Zinc (Zn), lead (Pb), and copper (Cu) dominate PC4 (6.8% variance), implying traffic emissions from non-exhaust sources and industrial processes. Erratic Zn peaks (e.g., 1.1% in Wet 2) are likely due to brake and tire wear from Medellín's dense vehicle fleet, while Pb points to industrial pollution. Although absolute concentrations are low (Zn: 0.089 $\mu$g/m$^3$), these contaminants have a high toxic potential even in small traces [70]. Finally, fluoride and chloride form PC5 (5.6% variance), indicating industrial influences. Chloride is unlikely to have a marine link in Medellín due to its landlocked geography; therefore, industrial processes (e.g., chemical manufacturing) are the likely source. Although minor (<1% of PM$_{2.5}$), their covariation ($\rho$ = 0.66) warrants monitoring given the toxicity of fluoride.

PC1, associated with crystalline material, contributes more to the total mass of PM$_{2.5}$ during dry periods due to increased dust resuspension and decreased precipitation. Conversely, the contribution of carbonaceous species and secondary aerosols (PC2) is greater during wet seasons because reduced solar radiation allows these compounds to remain in the atmosphere longer. The second dry season also coincides with events such as New Year's Eve celebrations, where fireworks increase the levels of specific metals in the pollutant composition.

During the study period, air quality in Medellín experienced distinct PM$_{2.5}$ peaks associated with extraordinary events. Wildfires were major contributors; in August 2019, smoke from large-scale fires in the Brazilian Amazon reached the

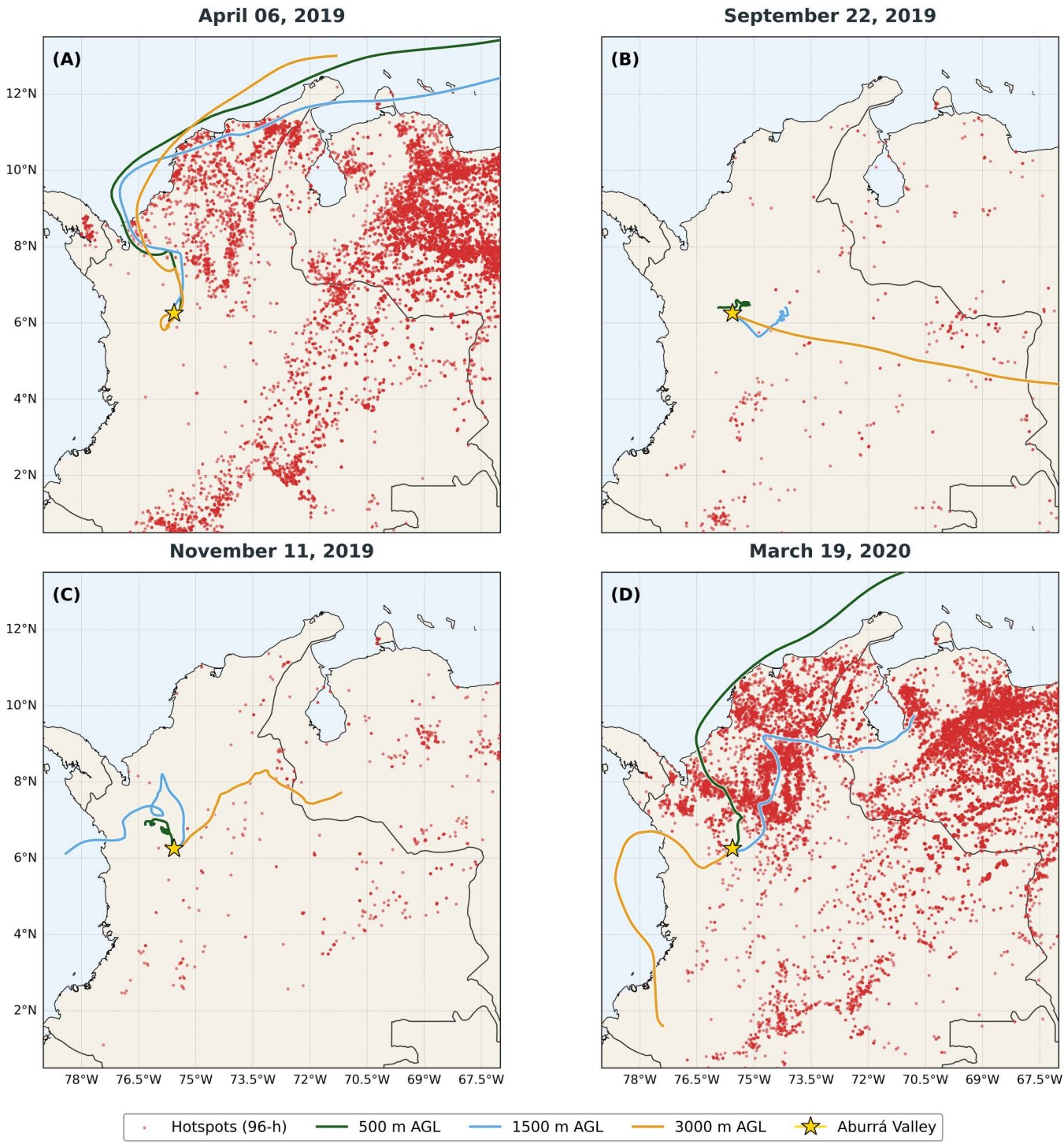

**Fig 8. Back-trajectories vs hotspots in the north of South America.** Integration of HYSPLIT backward trajectories and FIRMS hotspot data for four critical pollution events. The map illustrates the origin of air masses and their intersection with regional biomass burning areas characterized by high (**A** and **C**) and low (**B** and **D**) hotspot densities.

Table 4. Spearman correlation factor analysis among chemical species and $PM_{2.5}$.

| | $PM_{2.5}$ | $Na^+$ | $Mg^{2+}$ | Al | Si | $K^+$ | $Ca^{2+}$ | $F^-$ | $Cl^-$ | $NO_3^-$ | $SO_4^{2-}$ | OC | EC |
|---|---|---|---|---|---|---|---|---|---|---|---|---|---|
| $PM_{2.5}$ | 1 | – | – | – | – | – | – | – | – | – | – | – | – |
| $Na^+$ | 0.40 | 1 | – | – | – | – | – | – | – | – | – | – | – |
| $Mg^{2+}$ | 0.60 | 0.72 | 1 | – | – | – | – | – | – | – | – | – | – |
| Al | 0.68 | 0.58 | 0.77 | 1 | – | – | – | – | – | – | – | – | – |
| Si | 0.68 | 0.58 | 0.77 | 1.00 | 1 | – | – | – | – | – | – | – | – |
| $K^+$ | 0.60 | 0.20 | 0.40 | 0.41 | 0.41 | 1 | – | – | – | – | – | – | – |
| $Ca^{2+}$ | 0.51 | 0.58 | 0.76 | 0.78 | 0.78 | 0.28 | 1 | – | – | – | – | – | – |
| $F^-$ | 0.48 | 0.43 | 0.54 | 0.48 | 0.48 | 0.37 | 0.35 | 1 | – | – | – | – | – |
| $Cl^-$ | 0.62 | 0.46 | 0.47 | 0.46 | 0.46 | 0.38 | 0.38 | 0.66 | 1 | – | – | – | – |
| $NO_3^-$ | 0.55 | 0.11 | 0.12 | 0.14 | 0.14 | 0.47 | −0.03 | 0.32 | 0.45 | 1 | – | – | – |
| $SO_4^{2-}$ | 0.81 | 0.45 | 0.49 | 0.48 | 0.48 | 0.46 | 0.36 | 0.43 | 0.48 | 0.44 | 1 | – | – |
| OC | 0.82 | 0.41 | 0.40 | 0.39 | 0.39 | 0.65 | 0.25 | 0.49 | 0.53 | 0.58 | 0.61 | 1 | – |
| EC | 0.63 | 0.05 | 0.09 | 0.08 | 0.08 | 0.57 | −0.06 | 0.26 | 0.41 | 0.42 | 0.41 | 0.57 | 1 |

Correlation is significant at the 0.05 level (2-tailed). Bold values indicate strong correlation ($\rho > 0.7$). Values of 1.00 between Al and Si suggest a common crustal origin.

Table 5. Principal components loadings for identified $PM_{2.5}$ sources.

| Variable | PC1 | PC2 | PC3 | PC4 | PC5 |
|---|---|---|---|---|---|
| Na | 0.529 | – | – | – | 0.425 |
| Mg | 0.734 | – | – | – | – |
| Al | 0.953 | – | – | – | – |
| Si | 0.953 | – | – | – | – |
| Ca | 0.844 | – | – | – | – |
| Ti | 0.953 | – | – | – | – |
| Fe | 0.848 | – | – | – | – |
| Ba | 0.866 | – | – | – | – |
| $NO_3^-$ | – | 0.697 | – | – | – |
| $SO_4^{2-}$ | – | 0.844 | – | – | – |
| OC | – | 0.816 | – | – | – |
| EC | – | 0.686 | – | 0.443 | – |
| $Na^+$ | – | – | 0.567 | – | – |
| $K^+$ | – | – | 0.777 | – | – |
| $Mg^{2+}$ | – | – | 0.760 | – | – |
| $Ca^{2+}$ | – | – | 0.622 | – | – |
| Zn | – | – | – | 0.877 | – |
| Pb | – | – | – | 0.819 | – |
| Cu | – | – | – | 0.403 | – |
| $F^-$ | – | – | – | – | 0.802 |
| $Cl^-$ | – | 0.469 | – | – | 0.608 |

Extraction method: Principal Component Analysis. Rotation method: Varimax with Kaiser normalization. Only loading values over 0.4 are shown for clarity.

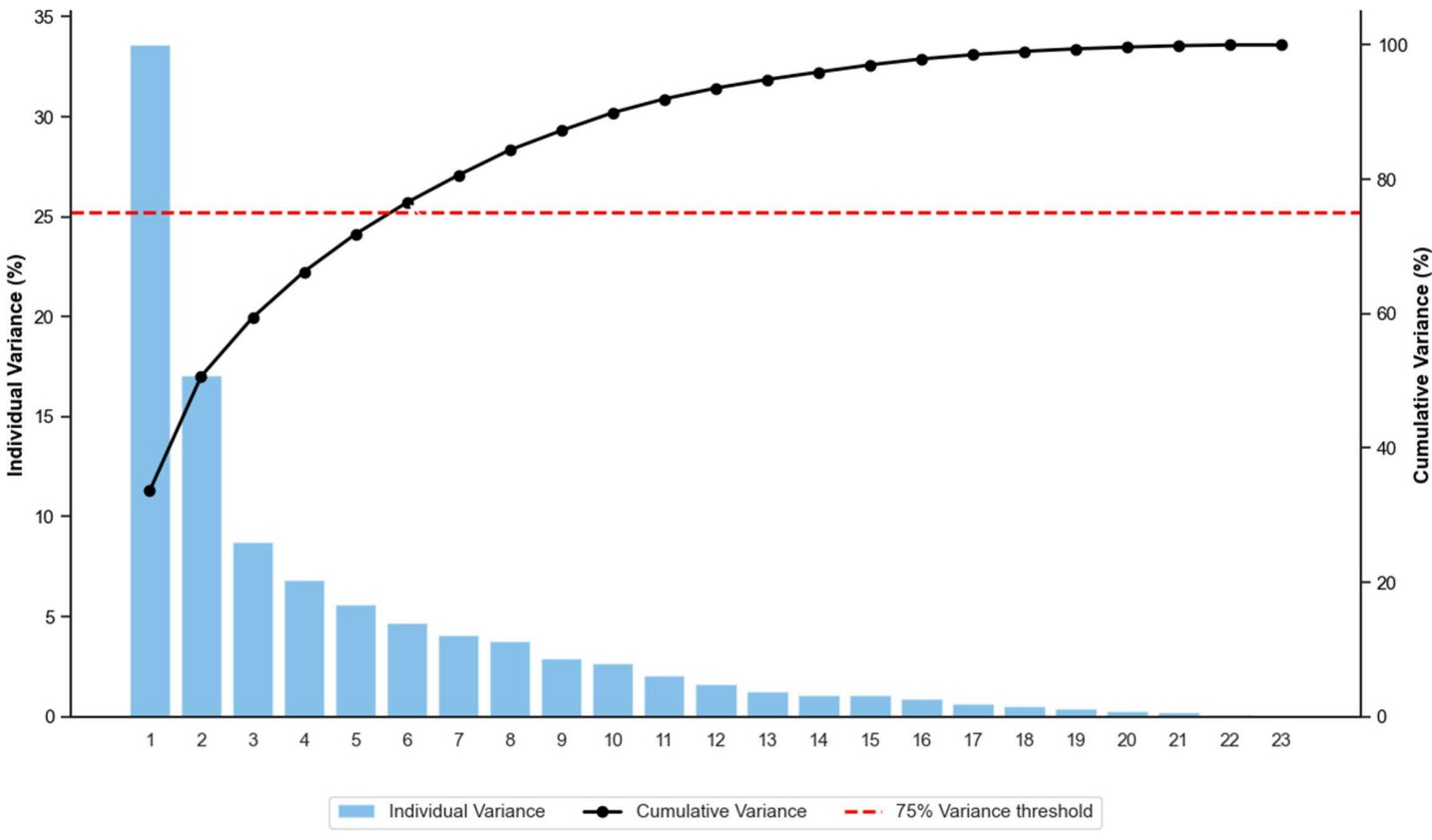

**Fig 9. Scree plot: Explained variance by principal component.** The plot displays the percentage of variance explained by each eigenvalue. The first five components exceed the 5% threshold, collectively accounting for more than 70% of the total cumulative variance in the chemical species dataset.

**Table 6. Detected sources through PCA.**

| PC | Variables | Identified Sources |
|---|---|---|
| PC1 | Na, Mg, Al, Si, Ca, Ti, Fe, Ba | Soil dust, crustal materials, road dust resuspension, construction activities |
| PC2 | $NO_3^-$, $SO_4^{2-}$, OC, EC | Secondary aerosol formation and mixed combustion |
| PC3 | K, $K^+$, $Mg^{2+}$, $Ca^{2+}$ | Biomass burning |
| PC4 | Zn, Pb, Cu | Traffic emissions (brake/tire wear), lubricants, industrial processes |
| PC5 | $F^-$, $Cl^-$ | Industrial emissions (e.g., fertilizers, aluminum production) |

Aburrá Valley, causing several monitoring stations to record levels unhealthy for sensitive groups. On the other hand, Christmas and New Year's Eve contributed to short-term spikes. Fireworks release large amounts of fine particulate matter, including metals and combustion products. Reports from environmental authorities show that on December 24, 25, 31, and January 1, $PM_{2.5}$ concentrations rose by as much as 160% compared to prior days. Immediately after detonations, levels can spike up to eight times the normal concentration. Similar behavior is assumed for the December 2019–January 2020 period in this study.

This aligns with the observation of record-high $PM_{2.5}$ levels in March 2020. The combination of dry conditions and widespread wildfires created an environment where pollutants accumulated to extreme levels. Conversely, the return of rainfall in April 2020 led to a dramatic drop in $PM_{2.5}$. The attribution of sources is complemented by the analysis of precipitation cycles and hotspot dynamics both inside and outside the Aburrá Valley, confirming that meteorological periods significantly impact the city's air quality.

## Conclusions

This study identified the composition and the main $PM_{2.5}$ sources at an urban background site in the Aburrá Valley, including mineral and resuspended dust, secondary aerosols, industrial emissions, traffic-related sources, and biomass burning. These contributors exhibit distinct temporal patterns throughout the year, governed by a complex interaction between local emissions, regional transport, and the valley's seasonal dynamics.

The identified patterns across the seasonal periods—ranging from high concentrations in wet seasons to the varying density of local and regional hotspots in dry seasons—demonstrate that the valley's air quality is governed by a shifting balance of internal accumulation and external transport. Furthermore, the transition between dry periods highlights a clear shift from the influence of local fire activity to a more significant impact of transboundary transport, particularly from the Magdalena River Basin.

Future research should focus on establishing statistical evidence regarding the influence of environmental variables on these pollutant concentrations. Additionally, future efforts should aim to integrate these chemical and source-specific profiles with public health and epidemiological data to evaluate the associated health risks and toxicological impacts on the valley's population. Utilizing these tracers and health-related datasets in future analytical frameworks will be essential to refine air quality management strategies and better protect the health of the population in topographically challenged urban environments.

## Supporting information

**S1 File. Replication scripts and data processing codes.** This compressed ZIP file contains all the R/Python scripts and processed datasets required to replicate the PCA analysis, HYSPLIT trajectories, and the statistical figures presented in this study.
(RAR)

## Author contributions

**Conceptualization:** Luisa M. Gómez Pelez, Santiago A. Franco.

**Data curation:** Miriam Gómez-Marín, Santiago A. Franco.

**Formal analysis:** Mauricio A. Correa-Ochoa, Miriam Gómez-Marín, Kelly Viviana Patiño-López.

**Funding acquisition:** Miriam Gómez-Marín.

**Investigation:** Mauricio A. Correa-Ochoa, Kelly Viviana Patiño-López.

**Methodology:** Kelly Viviana Patiño-López, Santiago A. Franco.

**Project administration:** Mauricio A. Correa-Ochoa, Miriam Gómez-Marín.

**Resources:** Mauricio A. Correa-Ochoa, Miriam Gómez-Marín.

**Software:** Mauricio A. Correa-Ochoa, Santiago A. Franco.

**Supervision:** Mauricio A. Correa-Ochoa, Luisa M. Gómez Pelez.

**Validation:** Luisa M. Gómez Pelez.

**Visualization:** Mauricio A. Correa-Ochoa, Santiago A. Franco.

**Writing – original draft:** Mauricio A. Correa-Ochoa, Miriam Gómez-Marín, Santiago A. Franco.

**Writing – review & editing:** Mauricio A. Correa-Ochoa, Miriam Gómez-Marín, Kelly Viviana Patiño-López, Luisa M. Gómez Pelez, Santiago A. Franco.

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
