## [Decision Letter · Decision Letter 0]

26 Nov 2025

PONE-D-25-58360Chemical Composition, Seasonal Variability, and Source Identification of PM2.5 at an Urban Background Site in Medellín—ColombiaPLOS ONE

Dear Dr. Correa Ochoa,

Thank you for submitting your manuscript to PLOS ONE. After careful consideration, we feel that it has merit but does not fully meet PLOS ONE’s publication criteria as it currently stands. Therefore, we invite you to submit a revised version of the manuscript that addresses the points raised during the review process. Please submit your revised manuscript by Jan 10 2026 11:59PM. If you will need more time than this to complete your revisions, please reply to this message or contact the journal office at plosone@plos.org. Please include the following items when submitting your revised manuscript:

We look forward to receiving your revised manuscript.

Kind regards,

Dipesh Rupakheti

Academic Editor

PLOS ONE

**Journal Requirements:**

1. When submitting your revision, we need you to address these additional requirements. Please ensure that your manuscript meets PLOS ONE's style requirements, including those for file naming. The PLOS ONE style templates can be found at https://journals.plos.org/plosone/s/file?id=wjVg/PLOSOne_formatting_sample_main_body.pdf and https://journals.plos.org/plosone/s/file?id=ba62/PLOSOne_formatting_sample_title_authors_affiliations.pdf 2. We noticed you have some minor occurrence of overlapping text with the following previous publication(s), which needs to be addressed: KINETICS OF UO2 DISSOLUTION UNDER HIGHLY ALKALINE CONDITIONS: APPLICATION OF A THIN FILM CONTINUOUSFLOW-THROUGH REACTOR-https://upcommons.upc.edu/server/api/core/bitstreams/914956f5-d608-4414-a995-c1661105f8e2/content Characteristic Chemical Profile of Particulate Matter (PM2.5)—A Comparative Study Between Two Periods, Case Study in Medellín, Colombia- https://doi.org/10.3390/su17125380 In your revision ensure you cite all your sources (including your own works), and quote or rephrase any duplicated text outside the methods section. Further consideration is dependent on these concerns being addressed. 3. In your Methods section, please provide additional information regarding the permits you obtained for the work. Please ensure you have included the full name of the authority that approved the field site access and, if no permits were required, a brief statement explaining why. 4. Thank you for stating the following financial disclosure: This research was funded by Ecopetrol (grant no. DHS 118) the Area Metropolitana del Valle de Aburrá (grant number 787)    Please state what role the funders took in the study.  If the funders had no role, please state: "The funders had no role in study design, data collection and analysis, decision to publish, or preparation of the manuscript." If this statement is not correct you must amend it as needed. Please include this amended Role of Funder statement in your cover letter; we will change the online submission form on your behalf. 5. We note that your Data Availability Statement is currently as follows: All relevant data are within the manuscript and its Supporting Information files. Please confirm at this time whether or not your submission contains all raw data required to replicate the results of your study. Authors must share the “minimal data set” for their submission. PLOS defines the minimal data set to consist of the data required to replicate all study findings reported in the article, as well as related metadata and methods (https://journals.plos.org/plosone/s/data-availability#loc-minimal-data-set-definition). For example, authors should submit the following data: - The values behind the means, standard deviations and other measures reported;- The values used to build graphs;- The points extracted from images for analysis. Authors do not need to submit their entire data set if only a portion of the data was used in the reported study. If your submission does not contain these data, please either upload them as Supporting Information files or deposit them to a stable, public repository and provide us with the relevant URLs, DOIs, or accession numbers. For a list of recommended repositories, please see https://journals.plos.org/plosone/s/recommended-repositories. If there are ethical or legal restrictions on sharing a de-identified data set, please explain them in detail (e.g., data contain potentially sensitive information, data are owned by a third-party organization, etc.) and who has imposed them (e.g., an ethics committee). Please also provide contact information for a data access committee, ethics committee, or other institutional body to which data requests may be sent. If data are owned by a third party, please indicate how others may request data access. 6. When completing the data availability statement of the submission form, you indicated that you will make your data available on acceptance. We strongly recommend all authors decide on a data sharing plan before acceptance, as the process can be lengthy and hold up publication timelines. Please note that, though access restrictions are acceptable now, your entire data will need to be made freely accessible if your manuscript is accepted for publication. This policy applies to all data except where public deposition would breach compliance with the protocol approved by your research ethics board. If you are unable to adhere to our open data policy, please kindly revise your statement to explain your reasoning and we will seek the editor's input on an exemption. Please be assured that, once you have provided your new statement, the assessment of your exemption will not hold up the peer review process. 7. We note that Figure 1 in your submission contain map images which may be copyrighted. All PLOS content is published under the Creative Commons Attribution License (CC BY 4.0), which means that the manuscript, images, and Supporting Information files will be freely available online, and any third party is permitted to access, download, copy, distribute, and use these materials in any way, even commercially, with proper attribution. For these reasons, we cannot publish previously copyrighted maps or satellite images created using proprietary data, such as Google software (Google Maps, Street View, and Earth). For more information, see our copyright guidelines: http://journals.plos.org/plosone/s/licenses-and-copyright. We require you to either present written permission from the copyright holder to publish these figures specifically under the CC BY 4.0 license, or remove the figures from your submission: a. You may seek permission from the original copyright holder of Figure 1 to publish the content specifically under the CC BY 4.0 license.   We recommend that you contact the original copyright holder with the Content Permission Form (http://journals.plos.org/plosone/s/file?id=7c09/content-permission-form.pdf) and the following text:“I request permission for the open-access journal PLOS ONE to publish XXX under the Creative Commons Attribution License (CCAL) CC BY 4.0 (http://creativecommons.org/licenses/by/4.0/). Please be aware that this license allows unrestricted use and distribution, even commercially, by third parties. Please reply and provide explicit written permission to publish XXX under a CC BY license and complete the attached form.” Please upload the completed Content Permission Form or other proof of granted permissions as an "Other" file with your submission. In the figure caption of the copyrighted figure, please include the following text: “Reprinted from [ref] under a CC BY license, with permission from [name of publisher], original copyright [original copyright year].” b. If you are unable to obtain permission from the original copyright holder to publish these figures under the CC BY 4.0 license or if the copyright holder’s requirements are incompatible with the CC BY 4.0 license, please either i) remove the figure or ii) supply a replacement figure that complies with the CC BY 4.0 license. Please check copyright information on all replacement figures and update the figure caption with source information. If applicable, please specify in the figure caption text when a figure is similar but not identical to the original image and is therefore for illustrative purposes only.The following resources for replacing copyrighted map figures may be helpful: USGS National Map Viewer (public domain): http://viewer.nationalmap.gov/viewer/The Gateway to Astronaut Photography of Earth (public domain): http://eol.jsc.nasa.gov/sseop/clickmap/Maps at the CIA (public domain): https://www.cia.gov/library/publications/the-world-factbook/index.html and https://www.cia.gov/library/publications/cia-maps-publications/index.htmlNASA Earth Observatory (public domain): http://earthobservatory.nasa.gov/Landsat: http://landsat.visibleearth.nasa.gov/USGS EROS (Earth Resources Observatory and Science (EROS) Center) (public domain): http://eros.usgs.gov/#Natural Earth (public domain): http://www.naturalearthdata.com/ 8. Please upload a new copy of Figures 2a and 2b as the detail is not clear. Please follow the link for more information:  https://journals.plos.org/plosone/s/figures 9. Please remove your figures from within your manuscript file, leaving only the individual TIFF/EPS image files, uploaded separately. These will be automatically included in the reviewers’ PDF. 10. If the reviewer comments include a recommendation to cite specific previously published works, please review and evaluate these publications to determine whether they are relevant and should be cited. There is no requirement to cite these works unless the editor has indicated otherwise.

Reviewers' comments:

Reviewer's Responses to Questions

**Comments to the Author**

1. Is the manuscript technically sound, and do the data support the conclusions?

Reviewer #1: Partly

Reviewer #2: Partly

Reviewer #3: Partly

2. Has the statistical analysis been performed appropriately and rigorously? 

Reviewer #1: N/A

Reviewer #2: Yes

Reviewer #3: Yes

3. Have the authors made all data underlying the findings in their manuscript fully available?

Reviewer #1: No

Reviewer #2: Yes

Reviewer #3: Yes

4. Is the manuscript presented in an intelligible fashion and written in standard English?

Reviewer #1: No

Reviewer #2: Yes

Reviewer #3: Yes

5. Review Comments to the Author

**Reviewer #1:** Some conclusions are not supported by the data. Eg, the traffic emissions contributions and their persistence. Also, it's difficult to check some conclusions as the months corresponding to PM2.5 concentrations are not mentioned in the figure.

Statistical analysis: As the authors have not made their data analysis code available, it's not possible to check the statistical analysis. However, just reading the methods, the statistical analysis looks correct.

Data availability - though the authors have mentioned that all the data are available without restriction, there is no reference to the raw data. There is no supporting information that contains the raw data or any online repository, eg., on GitHub.

English: The language of the paper can be improved.

Following is my feedback:

Overall, the authors collected substantial data to identify sources of PM₂.₅. However, the manuscript would benefit from improvements in writing clarity and more detailed descriptions of study area and meteorological conditions for each monitoring day. I recommend using a consistent definition of PC2 throughout the paper. Describing it solely as “secondary aerosols” in certain sections may be misleading to readers. In the discussion, it is unclear which specific months were affected by the COVID-19 lockdown. The manuscript states that lockdown began in mid-March 2020, but also mentions “mixed effects” beforehand without specifying the months involved.

Specific comments are provided below.

Abstract

Line 21: Change the capital “I” in “Industrial” to lowercase.

Line 22: Add standard deviations to the reported mean PM₂.₅ concentrations.

Lines 25- 27: Make it clear that traffic emissions here represents non-exhaust emissions. The conclusion is not well supported by the data. The so-called persistent sources, especially the non-exhaust traffic emissions only contribute <1% to total PM2.5 in all seasons (Table 6).

Introduction

Lines 84–86: Add references to support the statement about previous studies—what are the “few studies” referred to here?

Methods

Study Site

Provide more detail on surrounding roads: Are they fully paved, partially paved, or unpaved?

Clarify the sources of biomass burning in the region:

– Is it dominated by seasonal crop residue burning?

– Biomass use for cooking?

– Wood burning for heating?

Ensure consistent formatting of PM₂.₅ (use subscript for “2.5” throughout; e.g., lines 122, 127).

Line 120: Add the number of samples collected in each season.

Line 167: Correct to “analyzed”.

Results / Discussion

Line 187 (Table 2): Include standard deviations and medians for each variable measured in each season.

Why does the Dry 1 period have the lowest average concentration? Please discuss.

Figure 3:

– Mark rainy days or indicate the average rainfall on monitoring days, if any.

– Clarify the x-axis: include monitoring dates.

– Explain the smoothing method used for PM₂.₅ (e.g., CMA) and state the window size.

Line 255 (Table 5): In PC2, also include “mixed combustion”.

Line 262: Correct “PC” to “OC”.

Line 283: Table 6 is not cited in the text; please reference it.

Line 306: The effect of rainfall is not visible in the data. Need to mark dates in the figure 3.

Additionally, the impact of the lockdown is not apparent in the PM₂.₅ concentrations around mid-March 2020, mainly because the dates are not mentioned in the Figure 3.

**Reviewer #2:** General Comments

The manuscript titled ‘Chemical Composition, Seasonal Variability, and Source Identification of PM2.5 at an Urban Background Site in Medellín—Colombia’ explores the chemical composition and sources of PM in the city. The research is interesting and provides valuable insights. However, I think the manuscript needs improvements. Please check my comments below

• Section: 2.3 what is the error/uncertainty in the quantification of metals and ions

• Methodology section needs improvement in terms of what steps followed for quantification? What techniques used? What reagents used? This information will be helpful for readers to understand the research

• In table.2 please provide the SD along with mean. Also include the mean and SD of all the quantified chemical species

• PC5. Cl is also released from waste burning/mixed waste burning etc. How can you surely report it as Industrial emissions?

• I recommend authors to collect satellite VIIRS derived fire data and plot for the region around the city for the study period to show how many fire spots occur?

• It would be better if air mass back trajectories to the site is made and presented for the study period? May be concentration weighted trajectories?

• Provide some data on number of construction buildings in the city per year.

• Provide data of traffic vehicular count in the city

**Reviewer #3:** The manuscript entitled “Chemical Composition, Seasonal Variability, and Source Identification of PM2.5 at an Urban Background Site in Medellín – Colombia” analysed a year-long concentration of PM2.5 and its sources in Colombia. Even though the manuscript is written in a proper manner, it requires some significant changes that need to be implemented before publication. So a major revision is required to improve the quality. The General and specific comments are attached in seperate sheet.

6. PLOS authors have the option to publish the peer review history of their article (what does this mean?). If published, this will include your full peer review and any attached files.

Reviewer #1: No

Reviewer #2: **Yes:** vignesh prabhu

Reviewer #3: No

---

## [Author Response · Author response to Decision Letter 1]

7 Apr 2026

Dear Editor and Reviewers,

Thank you very much for the opportunity to revise our manuscript. We would like to express our sincere gratitude to the reviewers for their insightful comments and constructive suggestions, which have significantly helped us improve the quality and clarity of our work.

We have carefully addressed each of the points raised. In the following pages, we provide a point-by-point response to the reviewers' comments. For clarity, the reviewers' comments are presented in bold, followed by our detailed responses. Changes made in the revised manuscript have been highlighted to facilitate the review process (see track changes document).

General comments on the requested corrections:

Although the previous study "Characteristic chemical profile of PM2.5 particles: a comparative study between two periods. Case study in Medellín, Colombia" https://doi.org/10.3390/su17125380 contains a station contemplated in the present investigation, the previous one corresponds to the estimation of the characteristic profile of PM2.5 particles in two periods in different areas of the Aburrá Valley and nearby. In this study, the focus was on comparing the chemical compositions and the changes found between periods. The focus of the present investigation goes far beyond a temporal comparison, since several key aspects are analyzed, which we will now discuss: the analysis of climatic variability, in which the typical wet and dry periods are evaluated, as well as the biannual precipitation of the city of Medellín. This analysis is further detailed in the paper, since precipitation data were obtained from a meteorological station belonging to the main meteorological data entity of the country. This station is located one kilometer away from the point where the air quality station where PM2.5 samples were taken is located. Another important difference is the receiver model analysis (in this case, principal component analysis), which detected emission sources supported by multivariate statistics and the typical correlations between pollutants that are tracers of specific sources. Another important difference lies in the analysis of hot spots and back trajectories.

Regarding Kinetics of UO2 dissolution under highly alkaline conditions: Application of a thin film continuous flow-through reactor (https://upcommons.upc.edu/server/api/core/bitstreams/914956f5-d608-4414-a995-c1661105f8e2/content), we understand that there may be similarities, especially in the methodological part; however, this is because the methods are standardized and usually the same steps and descriptions are used to describe the procedures. In this case, we have made sure to improve the methodological section.

We do not wish to manifest changes in our sponsors or upload our procedures to protocols.io, as we have employed standardized methodologies widely used worldwide for the chemical analyses we have performed. We have also checked that all sources in both the preliminary and current versions of the document are correctly referenced, thus acknowledging the work of the scientific community around our topic of study.

We have added all the key documents for the preparation of the study in a ZIP folder, where a Jupyter notebook, programmed in Python, with the databases and algorithms that allow us to obtain the figures and data presented, is found as a fundamental input (Sup. File 1.zip). For this reason, we consider that our document contains all the information necessary for a scientist to replicate our findings. However, if someone wishes to request more information about the study, we offer the possibility of sharing more information upon formal request. We reserve the right of reply, as we leave all replication data available.

We emphasize in this section that our study did not include sensitive information involving restrictions due to ethical and/or legal reasons. With respect to figure 1, we have made changes in it to use images taken from free and open repositories, where it is not necessary to request licenses for use, since they work under the CC BY 4.0 license. To maintain consistency with the new graphic resources, we have updated figures 2 and 2b: the first one becomes figure 3 and we have omitted the second one, since with the new resources added it could be redundant. This resource can be found in our supplementary material. With this, we improved the graphic quality of the paper, as requested by the journal editor. We have removed all figures from the final document and attached them in TIFF format in the document upload platform. In the present study we have not found any suggestions for citations by the reviewers. We have added more information on permits obtained to work at the study site.

Response to Reviewer 1:

We have corrected the spot changes suggested by the reviewer regarding syntax and added the standard deviations to the pollutant concentrations for the overall period and for each of the periods. We have expanded the discussion of traffic emissions relative to internal city emissions and reported emissions per ton per year for this source. We have eliminated the table in which we presented the contribution of the component (PC) to the total mass of the pollutant, as the rigor of these calculations did not meet the requirements of the journal. We have contrasted this source information with the social and environmental characteristics that appear in this new version. In the introduction section, the reviewer states that the information is based on few previous studies. We refer to the fact that it is complex to find studies that quantify the relationships between pollutants and climate variables. For this reason, we do not make references, since this idea is based on the bibliographic search carried out in the framework of this research. To correct this ambiguity, we have changed the wording to make it clear what we are referring to. In the methodological section, we have added information on the context surrounding the study site.

In the discussion section, we have added more information on biomass burning and included an important part in the structure of the article consisting of hot spot analysis, supported with figures and back trajectory analysis. We leave for a future work the discretization of biomass sources, since in this paper we focus on what happens in the country and, more specifically, in the valley. We have placed Table 1 in the methodology section so that the discretization performed in the sampling and the number of samples taken in each of the periods are clearer.

In the discussion section, more information on biomass burning has been added and an important part of the article consists of hot spot analysis, supported with figures and back trajectory analysis. We leave for future work the discretization of biomass sources, since things happen in the country and, more specifically, in the valley. We have placed Table 1 in the methodology section so that the discretization performed in the sampling and the number of samples taken in each of the periods are clearer. We have expanded the discussion of why dry period 1 had the lowest concentrations of material. Due to the nature of the new plots included, we have employed a new library of plots so that everyone's style is standardized and the document has better aesthetics. We have taken into account the reviewer's suggestions and have added the dates on the x-axes of each of the graphs that require it. As we have done with the biomass, we have added relevant and contrasted information on precipitation periods, which are taken by the main meteorological institute of Colombia and are at a station located two kilometers from the measuring station. Regarding the pandemic issue, we chose not to include this information, since our study period does not cover the confinement extensively. Although it is public knowledge that in the Aburrá Valley the concentrations of particulate matter were at medium-high levels during the confinement, we have considered the relevance of this topic in relation to our research and, for this reason, we have decided not to include this information.

Response to Reviewer 2:

We have also added information on the uncertainties in the quantification methods. We have expanded the methodological information on our quantification techniques; however, we consider that this aspect may become redundant, since we have employed standardized methods widely used in similar studies, so they are usually referenced and known by the community. In addition, we have left references for each of these methods, which consist of extensive documents where the particular procedures for each research need are clearly explained. We also emphasize that these procedures have been carried out by laboratories recognized nationally by the main environmental studies entity in Colombia, which are governed by clearly established technical norms and, in many cases, comply with the standards of U.S. and European agencies. We understand that the internal management of each IDEAM-accredited laboratory may contain confidential protocols, so it is not possible to accurately describe each procedure. However, we adhere to the standardized protocol reported by the laboratory for the handling, quantification and quality of the reported data. We have added standard deviation data for the major contaminants for each of the periods studied. We have collected FIRMS data (VIIRS J2) to contrast the hot spots of the Colombian region and the Aburrá valley, finding important characteristics that support the arguments made in the first version. Similarly, we have performed an analysis of back trajectories, which we have studied with Hysplit at critical dates. We have also provided information on the city's traffic density and annual emissions, as well as on the city's industrial emissions.

Response to reviewer 3:

1. In many places, the authors used very generic statements that should be replaced with scientific terminology. This needs to be addressed throughout the manuscript.

We have improved the content of the manuscript by improving the wording and using more technical terms throughout the document.

2. The flow of the introduction needs to be re-examined. A few statements contradict others within the introduction and need to be addressed.

We do not consider there to be any contradictory ideas in the document; perhaps they were expressed incorrectly, which led the reviewer to reach this conclusion. We have worked hard to ensure that interpretations like this do not have a place in the current version of the document.

3. The introduction should explain how this study will be useful for further research by researchers or policymakers, and what new information it provides to existing scientific knowledge.

This aspect has been addressed explicitly in the discussion and conclusions, as it is an important aspect, since it is one of the purposes of the study and of science in general. We have added a section to the introduction to make readers aware of the importance of our document.

4. Section 2.1 should include information on local sources such as industries and transport corridors that surround the study area.

We have added more context about the area surrounding the study area.

5. Section 2.2 contains many abbreviations that need to be used consistently throughout the manuscript.

We have reviewed the abbreviations and corrected them where necessary.

6. Sections 2.3.1 to 2.3.3 can be combined into a single paragraph mentioning the methodology and procedure followed. These can be referred to in previous studies conducted by the authors and mentioned here with a reference instead of explaining the procedure again.

Other reviewers have requested a more detailed description of the methods, so we have decided to leave these sections as they were in the first version of the document.

7. The results section seems weak and could benefit from more rigorous scientific discussion. The authors should provide more analysis to illustrate the differences in concentration between the wet 1, wet 2, dry 1 and dry 2 seasons. Interannual changes.

We have strengthened our results by supporting them with new information on hotspots, precipitation and back-trajectories in the study area, and expanded the discussion on what happens in each of the periods of our study.

8. The authors should investigate the different mean and sum concentrations mentioned in the manuscript. There are numerous mismatches in mean and sum values throughout the manuscript.

Very important aspect highlighted by the reviewer. We appreciate the thorough review of the document. We have reanalyzed the information to avoid making this type of error. We have solved the inconsistencies and are working firmly on a single dataset, which is attached to the document and contains the information so that the experimentation can be replicated by anyone with basic knowledge of Python and R Studio.

9. Figures 2 and 3,4 are not properly labelled. Only the unit is written. What do they represent, e.g. mass concentration, and what is being counted? This needs to be updated.

We have significantly improved the presentation of our graphs throughout the document.

10. The discussion sections need significant improvement. The section contains redundant sentences and the information is not discussed scientifically.

We have worked hard to improve the language in our document, and we have discussed and supported the information with our analyses and the data reported by entities such as NOAA and NASA, as well as IDEAM. We have also contrasted the information with solid bibliographic references to widely referenced and published studies in recognised and cutting-edge journals in the field of research.

Specific comments:

1. On line 32, the authors used generic statements instead of specific scientific terminology. For example, 'within cities' could be replaced with 'local sources'.

This has been corrected.

2. Similarly, on line 33, 'emission generated miles away' was replaced with 'long-range transportation of sources'.

We have corrected this.

3. On line 34, PM2.5 is referred to as 'aerosols'. In most cases, aerosols are referred to as two smaller particles.

We have changed this.

4. Lines 41–42 remain as 'deposited'.

We have corrected this.

5. Line 42. The sentence contradicts the previous statement.

We do not consider the sentences to contradict each other; however, we have changed the wording so that the idea is clearer.

6. Lines 60 to 70 could be revised to provide basic information about the study area and the results of previous studies instead of CCN, since the impacts of CCN are more significant with particles in the smaller size range.

We agree with the reviewer's observation, so we have removed the idea related to CCN. Throughout the introduction and discussion, references are made to studies conducted in Medellín and Colombia. We have not delved into this aspect for this specific section.

7. Lines 78 to 79 provide information on how the concentration increased over time in Medellín.

We have discussed this in other sections of the document and do not consider it pertinent to include it in this specific section.

8. Line no. 95: Check one temperature unit.

Correction made.

9. Lines 98 to 100 do not match the information provided in lines 180. Cross-check and correct.

We have reviewed the additional information and verified its consistency. We have moved Table 1 to the Methodology section to make it more consistent with what we want to show.

10. If possible, replace Figure 1 with an image showing the surrounding Ares road network instead of the DEM map.

We have replaced Figure 1 in accordance with the style of the journal and the reviewer's suggestions.

11. On line 115, the authors should clarify whether the filter papers were desiccated before and after the measurement to remove moisture. Moisture content plays a major role in the chemical composition of the collected sample. If not preserved properly, the composition of the measured pollutants will change.

We have specified our procedures; however, we would like to emphasise that these procedures were carried out by a laboratory that follows standardised procedures issued by European and US environmental agencies, a

---

## [Decision Letter · Decision Letter 1]

12 May 2026

Chemical Composition, Seasonal Variability, and Source Identification of PM2.5 at an Urban Background Site in Medellín—Colombia

PONE-D-25-58360R1

Dear Dr. Correa Ochoa,

We’re pleased to inform you that your manuscript has been judged scientifically suitable for publication and will be formally accepted for publication once it meets all outstanding technical requirements.

Kind regards,

Dipesh Rupakheti

Academic Editor

PLOS One

Additional Editor Comments (optional):

Reviewers' comments:

Reviewer's Responses to Questions

**Comments to the Author**

1. If the authors have adequately addressed your comments raised in a previous round of review and you feel that this manuscript is now acceptable for publication, you may indicate that here to bypass the “Comments to the Author” section, enter your conflict of interest statement in the “Confidential to Editor” section, and submit your "Accept" recommendation.

Reviewer #1: All comments have been addressed

Reviewer #2: All comments have been addressed

Reviewer #3: All comments have been addressed

2. Is the manuscript technically sound, and do the data support the conclusions?

Reviewer #1: Yes

Reviewer #2: Partly

Reviewer #3: Yes

3. Has the statistical analysis been performed appropriately and rigorously? 

Reviewer #1: Yes

Reviewer #2: Yes

Reviewer #3: Yes

4. Have the authors made all data underlying the findings in their manuscript fully available?

Reviewer #1: Yes

Reviewer #2: Yes

Reviewer #3: Yes

5. Is the manuscript presented in an intelligible fashion and written in standard English?

Reviewer #1: Yes

Reviewer #2: Yes

Reviewer #3: Yes

6. Review Comments to the Author

Reviewer #1: (No Response)

Reviewer #2: (No Response)

Reviewer #3: The authors had put a significant effort in correcting the manuscript. Now the manuscript can be considered for publication

7. PLOS authors have the option to publish the peer review history of their article (what does this mean?). If published, this will include your full peer review and any attached files.

Reviewer #1: No

Reviewer #2: **Yes:** vignesh prabhu

Reviewer #3: No

---

## [Editor Report · Acceptance letter]

PONE-D-25-58360R1

PLOS One

Dear Dr. Correa-Ochoa,

I'm pleased to inform you that your manuscript has been deemed suitable for publication in PLOS One. Congratulations! Your manuscript is now being handed over to our production team.

Kind regards,

on behalf of

Dr. Dipesh Rupakheti

Academic Editor

PLOS One